# MiRNA Profiling in Plasma and Placenta of SARS-CoV-2-Infected Pregnant Women

**DOI:** 10.3390/cells10071788

**Published:** 2021-07-15

**Authors:** Irma Saulle, Micaela Garziano, Claudio Fenizia, Gioia Cappelletti, Francesca Parisi, Mario Clerici, Irene Cetin, Valeria Savasi, Mara Biasin

**Affiliations:** 1Department of Biomedical and Clinical Sciences, University of Milan, 20157 Milan, Italy; irma.saulle@unimi.it (I.S.); micaela.garziano@unimi.it (M.G.); claudio.fenizia@unimi.it (C.F.); gioia.cappelletti@unimi.it (G.C.); irene.cetin@unimi.it (I.C.); valeria.savasi@unimi.it (V.S.); 2Department of Pathophysiology and Transplantation, University of Milan, 20122 Milan, Italy; mario.clerici@unimi.it; 3Unit of Obstetrics and Gynecology, ASST Fatebenefratelli-Sacco, Department of Biological and Clinical Sciences L. Sacco, University of Milan, 20157 Milan, Italy; francesca.parisi288@gmail.com (F.P.); 4IRCCS Fondazione Don Carlo Gnocchi, 20148 Milan, Italy; 5Department of Woman, Mother and Neonate Buzzi Children’s Hospital, ASST Fatebenefratelli-Sacco, 20157 Milan, Italy

**Keywords:** SARS-CoV-2, pregnant women, MicroRNA, immune response

## Abstract

MicroRNAs are gene expression regulators associated with several human pathologies, including those generated by viral infections. Their role in SARS-CoV-2 infection and COVID-19 has been investigated and reviewed in many informative studies; however, a thorough miRNA outline in SARS-CoV-2-infected pregnant women (SIPW), at both systemic and placental levels, is missing. To fill this gap, blood and placenta biopsies collected at delivery from 15 asymptomatic SIPW were immediately analysed for: miRNA expression (*n* = 84) (QPCR array), antiviral/immune mRNA target expression (*n* = 74) (QGene) and cytokine/chemokines production (*n* = 27) (Multiplex ELISA). By comparing these results with those obtained from six uninfected pregnant women (UPW), we observed that, following SARS-CoV-2 infection, the transcriptomic profile of pregnant women is significantly altered in different anatomical districts, even in the absence of clinical symptoms and vertical transmission. This characteristic combination of miRNA and antiviral/immune factors seems to control both the infection and the dysfunctional immune reaction, thus representing a positive correlate of protection and a potential therapeutic target against SARS-CoV-2.

## 1. Introduction

Despite the high number of SARS-CoV-2-infected individuals worldwide, only a few studies have focused on pregnant women, and a number of questions concerning the molecular mechanisms controlling SARS-CoV-2 in in utero transmission are still unanswered.

The placenta has evolved several first-line strategies to actively defend the foetus against pathogens and to prevent vertical transmission, including the peculiarities of its architecture [1,2] and the activation of fine-regulated immune defence pathways [3]. Nevertheless, several viruses, bacteria, protozoa, and fungi can cross the transplacental barrier and disturb normal foetal development, possibly resulting in spontaneous abortion or premature birth [4,5].

In utero transmission of SARS-CoV-2 has been properly documented [6,7]. In these cases, a generalized hyper-activation of the immune response has been observed at both systemic and placental level, suggesting that an alteration of the immunological barrier may be responsible for in utero mother-to-child SARS-CoV-2 infection [6,7]. The orchestration of this immunological barrier depends on a number of molecular mechanisms, among which microRNAs (miRNAs) have recently emerged as playing a leading role [8].

MicroRNAs (miRNAs) are small endogenous noncoding molecules (21–25 nt length) that govern gene expression by mRNA degradation or by interfering with translation, depending upon their 3′UTR complementarity [9]. MiRNAs can regulate various biological and pathological pathways, including the replicative cycle and pathogenesis of RNA viruses, through both direct andindirect mechanisms [10]. Actually, viral infections can alter the host miRNA expression; additionally, these alterations can impair signalling pathways, modulate host–virus interactions, regulate viral infectivity and transmission, and result in the differential activation of antiviral immune responses [11,12].

Different studies have investigated the role of host miRNAs in SARS-CoV-2 infection, through both computational prediction analysis andexpression levels of miRNAs in biological specimens of COVID-19 patients [13]. The results have shown that altered miRNA levels in different COVID-19 cohorts could reflect distinct phases of the host antiviral immune response, from disease onset to recovery [14]. Such effects would stem from the double function exerted by miRNAs in the infected cell, by controlling the virus infection and/or regulating the immunity-related gene targets through complex networks of virus–host cell interactions [15,16].

A thorough investigation of the miRNA profile in SARS-CoV-2-infected pregnant women (SIPW) at both the systemic and local level, primarily in placenta samples, is missing; such information could contribute to the identification of targets for the treatment of COVID-19, and provide non-invasive biomarkers for COVID-19 diagnosis in this special population.

Based on these premises, we analysed the expression of 84 miRNAs and 74 mRNA with known antiviral and/or immunological activity in blood and human placenta biopsies from SIPW and uninfected pregnant women (UPW) (Figure 1), so as to:

Verify that miRNA expression in SIPW differs from that of UPW;

Assess if there is a correspondence between plasma and placenta miRNA profile;

Establish a possible correlation between miRNAs and the immunological profile, assessed through the evaluation of a panel of antiviral/immunological markers.

## 2. Methods

### 2.1. Study Population

This is a prospective study that includes 21 women: 6 uninfected pregnant women (UPW) and 15 SARS-CoV-2-infected pregnant women (SIPW) at first diagnosis, who were admitted at delivery in L. Sacco COVID-19 maternity hospital in Lombardy, Italy, between March and November 2020. The uninfected women are the control group. An in vitro SARS-CoV-2 neutralization assay confirmed that none of the UPW were infected by SARS-CoV-2 (data not shown).

All women underwent clinical evaluation of vital signs and symptoms, laboratory analysis and radiological chest assessment on admission, at the discretion of physicians. Consequently, the therapeutic management was tailored according to the clinical findings and national guidelines. Demographic and anthropometric characteristics and medical and obstetric comorbidities were recorded at enrolment through a customized data collection form. All pregnancies were singleton, with a normal course and regular checks, until delivery. All the women were admitted to a reference hospital for COVID-19 because they had to give birth.

The protocol was approved by the local Medical Ethical and Institutional Review Board (Milan, area 1, #154082020). Informed consent was obtained from all subjects involved in the study, in compliance with the Declaration of Helsinki principles.

### 2.2. Plasma, Peripheral Blood Mononuclear Cells (PBMCs) and Placenta Biopsy Collection

A few hours before delivery, a 10 mL maternal blood sample in EDTA was collected. Soon after delivery, full-thickness placental biopsies were obtained by a dedicated operator. Prior to the biopsy being taken, the placenta was delicately washed by a sterile physiologist to remove any maternal blood. In order to preserve RNA stability, biological samples from obstetrics and gynaecology units were immediately conveyed to the laboratory of immunology, University of Milan, to be processed or stored.

Plasma was obtained by centrifugation of whole blood, while basal PBMCs were separated on Lympholyte-H separation medium (OrganonTeknica; Malvern, PA, USA), as previously described [17].

### 2.3. Total RNA and miRNA Extraction from Placenta Biopsies, PBMCs and Plasma

Total RNA was extracted from placenta and PBMCs, as previously described [17]. Total MicroRNAs were isolated from 600 μL of plasma by NucleoSpinMiRNA Plasma Kit (MACHEREY-NAGEL, Düren, Germany), in accordance with the manufacturer’s protocol.

### 2.4. MicroRNA Reverse Transcription and Real-Time PCR Array Analysis

One microgram of RNA, isolated from plasma and placenta biopsies, was reverse-transcribed into first-strand cDNA in a 20 μL final volume at 37 °C for 60 min using miScript II RT Kit (Qiagen, Venlo, The Netherlands), in accordance with the manufacturer’s protocol. Expression level of 84 miRNAs with antiviral and/or immunological function was evaluated using a miRNA PCR Array (MIHS-111Z) (Qiagen, Venlo, The Netherlands). Experiments were run on all women included in the study. The arrays were performed on CFX ConnectTM Real-Time PCR system (BIO RAD, Hercules, CA, USA). Undetermined raw CT values were set to 35. Expression profile was analysed using the PCR Array Gene Expression Analysis Software (SABiosciences, Frederick, MD, USA). For each miRNA, Ct values were transformed into relative quantities using as a normalization factor, the arithmetical mean of the references available in the arrays, RNU6-2 for placenta and Cel_miR-39_1, miR-93 and RNU6-2 for plasma. Fold regulation of ±2.5 was considered as positive.

### 2.5. Quantigene Plex Gene Expression Assay

A total of 100 ng RNA extracted from PBMCs and placenta biopsies was used for gene expression analyses by QuantiGene Plex assay (Thermo Scientific, Waltham, MA, USA). This approach provides a fast and high-throughput solution for multiplex gene expression quantitation, allowing for the simultaneous measurement of 74 custom-selected genes of interest in a single well of a 96-well plate. The QuantiGene Plex assay is hybridization-based and incorporates branched DNA technology, which uses signal amplification for direct measurement of RNA transcripts. Results were calculated relative to GAPDH, β-Actin and PPIB as housekeeping genes, and expressed as ΔCt.

### 2.6. Cytokine and Chemokine Measurement by Multiplex Assay

Concentration of 27 cytokines/chemokines was assessed in the plasma of all women enrolled in the study by using immunoassays formatted on magnetic beads (Bio-rad, Hercules, CA, USA), according to manufacturer’s protocol, via Luminex 100 technology (Luminex, Austin, TX, USA). Some of the targets resulted were over-range and an arbitrary value of 4000 pg/mL was assigned, while 0 pg/mL was attributed to values below the limit of detection.

### 2.7. Statistical Analyses

For the study, media ± error standards were reported for quantitative variables. Biomarkers were compared between SIPW and UPW using Student’s *t*-test or Mann–Whitney U test for continuous variables. After performing log transformation of continuous variables to approximate to normal distribution, multivariate linear regression models were performed to investigate associations between maternal infection and the study variables’ concentrations, taking the following confounding factors into account: timing of infection, severity of infection and route of delivery. Data were analyzed using GRAPHPAD PRISM version 8 (Graphpad software, La Jolla, CA, USA), and *p*-values of 0.05 or less were considered significant.

## 3. Results

### 3.1. Population

The main features of the study population are summarized in Table 1 and Table 2.

The baseline characteristics of the two groups were similar. Maternal median age was 32 years (range 21–39) in SIPW and 33.5 years (range: 28–40 years) in UPW. Median pre-pregnancy BMI was normal and comparable in both groups (23.4 vs. 22.9). None of the patients had a smoking habit and the majority had no chronic comorbidity. Notably, by analysing the white blood cell count, routinely performed just before delivery (Appendix A), we observed that the lymphocyte cell count was significantly lower in SIPW compared to UPW (data not shown).

There were no significant differences in outcomes for newborns. Median birth weight was 3160 g (range: 2665–3775 g) in SIPW; in UPW, median birth weight was 2955 g (range: 2715–3500 g). In each group, there was a newborn with an APGAR score less than 7 at 5′. None of the newborns were SARS-CoV-2-infected and none of them entered NICU. Median umbilical artery pH was 7.34 (range: 7.24–7.53) in SIPW and 7.31 (range 7.19–7.36) in UPW.

All the placentas, both from cases and controls, were stored and analysed at the Pathology Unit in L. Sacco Hospital. We did not observe relevant differences in placenta histopathological patterns between SIPW and UPW.

### 3.2. MiRNA Expression in Human Plasma from SIPW and UPW

The expression of 84 miRNAs exerting an immunomodulant and/or antiviral function was assessed by real-time PCR in plasma of 6 UPW and 15 SIPW. The results showed that, of the 84 miRNAs, 71 showed no significant difference between the groups, while 35 displayed a differential expression level in SIPW compared to UPW. Notably, seven antiviral miRNAs (miR-21, miR-23b, miR-28, miR-29a, miR-29c, miR-98 and miR-326) (Figure 2A) and six immunomodulatory miRNAs (miR-17, miR-92, miR-146, miR-150, miR-155, miR-223) (Figure 2B) were upregulated in SIPW compared to UPW. These molecules include the non-coding molecules predicted to directly bind SARS-CoV-2 transcripts RNAs (miR-21b, miR-29c, miR-98), as well as miRNAs targeting cellular factors that indirectly influence SARS-CoV-2 replication and immune response (miR-146a, miR-150 or miR-155). Statistical significances are displayed in Figure 2A,B.

### 3.3. MiRNA Expression in Human Placenta Biopsies from SIPW and UPW

The expression levels of the same miRNAs were measured in RNA isolated from human placenta biopsies collected from SARS-CoV-2-infected and uninfected women. The expression of 76 miRNAs was comparable in the two groups, yet the synthesis of eight miRNAs significantly differed between SIPW and UPW. In particular, we observed an upregulation of miRNAs acting on viral replication through both direct (miR-21b, miR-29c, miR-98) (Figure 2C) and indirect mechanisms (miR-146, miR-155, miR-190, miR-346, and miR-326) (Figure 2D). Many of the miRNAs produced at the placental level in SIPW mirror the ones released in the blood flow, validating the assumption that their production is virus-specific and that they could be used as biomarkers to monitor viral infection. Statistical significances are reported in Figure 2C,D.

### 3.4. Gene Expression of Immune/Antiviral Selected Effectors in PBMCs and Placenta Biopsies from SIPW and UPW

The mRNA expression of 74 selected antiviral and immunomodulatory effectors was further investigated in RNA from placenta biopsies and PBMCs isolated from 6 UPW and 15 SIPW. The gene expression pattern was partially shared in the two anatomical districts, and was characterized by a slight increase in the expression of activation markers (CD69, CD38), chemokines (CCL3, CCL5), pro-inflammatory (IL-6, IL-1β, IFNα, IFNβ) and anti-inflammatory cytokines (IL-10) in SIPW compared to UPW, although these differences were far more evident at the systemic level (Figure 3A,B). Notably, we also detected a significant increase in the expression of host antiviral effector genes, such as MX1, IFITM1, IFITM3, and CH25H, in SIPW, mainly at the placenta level (Figure 3A,B).

One of the probes in the array was specific for SARS-CoV-2 genome, but neither placental nor PBMC specimens tested positive. Statistical significances are displayed in Figure 3, panel A and B.

These results suggest that, following SARS-CoV-2 infection, the immune response is activated at both systemic and local levels, but this activation is somehow kept under control to prevent degeneration into a hyper-activation status.

### 3.5. Modulation of Cytokine and Chemokine Production in Plasma from SIPW and UPW

Next, to elucidate the antiviral/immunomodulatory outline conveying the expression of the identified miRNAs, we assessed the production of 27 cytokines and chemokines related to immune response activation in plasma samples of infected/uninfected SARS-CoV-2 pregnant women. Similar to what was observed when gene expression was analyzed, a trend towards an increase in the inflammatory profile was seen in plasma samples of SIPW compared to those of UPW (Figure 4). In particular, we observed an increase in the secretion of pro-inflammatory cytokines, including IL-1β, IL-6, IL-10, IFNγ, TNFα, as well as in the CCL3 and CCL5 chemokines in plasma from SIPW. These data suggest that peripheral level SARS-CoV-2 infection in SIPW is characterized by an increased production of several cytokines, even in the absence of evident clinical symptoms.

## 4. Discussion

While SARS-CoV-2/host miRNA interactions have been carefully investigated in cohorts with a different clinical outcome or through in silico prediction analysis [13,14,18,19,20,21,22], only a limited number of analyses focused on the miRNA/transcriptional profile of SARS-CoV-2-infected pregnant women. Difficulties related to the proper collection and storage of biopsy specimens from women in labor, for further bio-molecular analyses, may have slowed down the gathering of such essential information. Significantly, data from this study are the first to describe the miRNA/transcriptional events occurring in SIPW. Their relevance is inferred by comparison with state-of-the-art knowledge on other SARS-CoV-2 cohorts and/or other viral infections at both systemic and placental levels.

The results showed that SARS-CoV-2 infection in SIPW induces the upregulation of miRNAs (miR-29a-29c-21-98), whose expression is known to directly and indirectly modulate viral infections, including SARS-CoV-2 [12,23,24,25,26,27]. In particular, in a recent in silico study, the miR-29 family showed the greatest number of interactions (11 sites) with the SARS-CoV-2 transcripts, primarily in the spike and nucleo-capsid coding regions [20]. Moreover, two bioinformatic analyses reported a possible correlation between miR-21 expression and a direct effect on SARS-CoV-2 genome, pinpointing four predictive binding sites [21,22]. Hence, the rise in biological specimens from SIPW resembles a defensive attempt of the organism to interfere with SARS-CoV-2 replication by directly targeting its gene expression, even at the placenta level. Concurrently, these same miRNAs may hamper viral infection/spreading by controlling the gene expression of host factors partaking in the SARS-CoV-2 viral cycle. Through a bioinformatics approach, Matarese et al. identified miR-98 as a suitable candidate regulator for TMPRSS2 transcription [28]; therefore, its increase in placenta biopsies from SIPW could contribute to reducing the risk of infection and, in turn, vertical transmission. MiR-21, instead, could exert an indirect effect on viral replication through its interaction with multiple host transcripts [22,24,27,29], mainly CXCL10, whose increased expression has been associated with the onset of a cytokine storm and a worst prognosis [30,31].

SARS-CoV-2 infection in our cohort was also accompanied by the upregulation of some miRNAs known to orchestrate the innate and adaptive immune response: miR-146, miR-150, and miR-155. The role of miR-146 in attenuating inflammation and natural killer (NK) degranulation is widely recognised in several pathological conditions, [29,32,33,34,35], as well as in other viral infections [36]. It is, therefore, plausible that miR-146 rise in SIPW might contribute to the control of severe COVID-19 by limiting inflammatory damage of tissues, as well as by negatively regulating NK-dependent cytotoxicity. In this study, for the first time, we also reported an increased expression of miR-150 following SARS-CoV-2 infection. As with any in silico study that has established a possible interaction between miR-150 and SARS-CoV-2 genome, we believe that its role in SIPW is to contain the hyperactivation of the immune system. Indeed, it is recognised that miR-150 is able to negatively regulate SOCS1, a suppressor of cytokine signalling whose aberrant expression could lead to cytokine dysregulation, as reported in dengue patients [37]. As for miR-155, its production has been associated with the modulation of various viral infections [38,39,40], including SARS-CoV-2 one, in in vitro infected cell lines [41]. Its real function in the control of the infection is uncertain: on one hand, miR-155 could play a protective role via CD4+ T cell regulation and by affecting CD8+ T cell responses [42,43]; on the other hand, lung injury by acute respiratory distress syndrome (ARDS) was attenuated by the deletion of miR-155, making this miRNA a potentially harmful player in the context of COVID-19. Further analyses are needed to ascertain the role of miR-155 in SARS-CoV-2 infection, although its upregulation in asymptomatic SIPW let us to speculate on a possible shielding function.

Even more importantly, we cannot forget that the expression of almost all of these miRNAs has also been conveyed in pregnancy-associated disorders. For example, miR-21, miR-29, miR-150 and miR-155 are highly expressed in human placentae of preeclampsia patients [44,45,46]; aberrant expression of miR-21 was reported to be associated with foetal hypoxia, foetal growth restriction, and macrosomia [47,48,49]; mir-29, miR-98 were upregulated in placentas from gestational diabetes mellitus patients [50,51]. For all these reasons, although the induction of these miRNAs has been associated with an effective control of viral replication, and no SIPW enrolled in this study had obstetric pathology related to placental malfunction, we cannot exclude that the alteration of miRNA profile might result from pregnancy-related complications.

The synthesis of these miRNAs coupled with an immune profile in SIPW is characterized by the release of pro-inflammatory cytokines/chemokines (IL-6, IL-1β, CCL5, and CCL3) and type I interferons (IFNα and IFNβ) in both anatomical districts. These data were further confirmed by those obtained by the Luminex assay on plasma samples, and mirror the outline distinctive of subjects with reduced susceptibility to other viral infections. In HIV-1 infection, for example, natural resistance to viral exposure was repeatedly associated with the generation of an immune activation profile, in terms of both miRNA and pro-inflammatory cytokine release, which, under controlled conditions, were able to exert a protective effect [14,25,52]. Endorsing this speculation, the increased expression of a type I interferon in SIPW resulted in the downstream establishment of an antiviral state prompted by the transcription of several interferon-stimulated genes (ISGs): IFITM1, IFITM3, MX1 and CH25H. Their antiviral mechanism of action has already been documented in several microbial infections [53,54], as well as in SARS-CoV-2 one [52,55,56,57]. It is, therefore, plausible that they might contribute to placenta infection and vertical transmission in our SIPW cohort. However, as already reasoned for miRNA release, it will be necessary to consider the pros and cons of their induction at the placenta level. Indeed, like other oxysterols, 25-hydroxycholesterol (25HC), the enzymatic product of CH25H, seems to play a negative role in this anatomical compartment by triggering a proinflammatory microenvironment, and inhibiting the differentiation and fusion of term primary trophoblasts [58]. Likewise, IFITM expression at the placenta level has been associated with the inhibition of syncytiotrophoblast formation and foetal demise in two recently published studies [59,60]. Hence, it will be necessary to verify if the antiviral effect exerted by these ISGs, following SARS-CoV-2 infection, may accidentally result in a worse pregnancy outcome.

Overall this manuscript shows that an altered miRNA and gene expression profile may be elicited in different anatomical districts from SIPW, even in the absence of clinical symptoms and vertical transmission (as defined by negative SARS-CoV-2 RT-PCR test).

The scarce number of subjects enrolled, mainly those with a severe form of the disease, and the lack of functional assays correlating miRNA/transcript profiles, constitute limits to this study that need to be addressed by further focused analyses. Moreover, we cannot exclude that the differences reported in miRNA/mRNA expression levels distinguishing SIPW from UPW rely on the reduced lymphocyte cell count of SIPW. However, even under such circumstances, this should be considered as an indirect consequence of SARS-CoV-2 infection. Indeed, lymphonia has been reported as a marker of COVID-19 since the beginning of the pandemic [61].

Although preliminary, these results are the first ones portraying the miRNA/transcriptional outline of SIPW in different anatomical compartments and could provide key information regarding the strategies adopted by the human organism to prevent the worsening of clinical symptoms and vertical transmission. Overall, the induction of pro-inflammatory and antiviral effectors registered in our cohort, in response to SARS-CoV-2 entry, seems to be properly counterbalanced by the synthesis of anti-inflammatory cytokines (IL-10) and antiviral/immunomodulant non-coding RNA that are able to “fine-tune” the immune response into an appropriate range. Ultimately, this peculiar mixture of transcribed factors seems to contain both the infection and the dysfunctional immune response, which sometimes occurs in COVID-19 patients, thus representing a positive correlate of protection and a possible therapeutic target against SARS-CoV-2.

## Figures and Tables

**Figure 1 cells-10-01788-f001:**
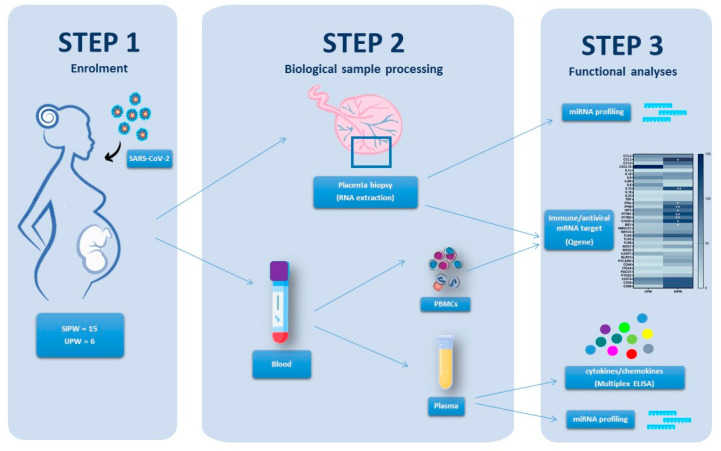
Graphical representation of the study workflow. Fifteen-SIPW and 6 UPW were enrolled in the study (**Step1**). At delivery, blood and placenta biopsies were collected from obstetrics and gynaecology units and immediately conveyed to the laboratory of immunology to be processed for further analyses (**Step 2**). Analyses of miRNA (*n* = 84), antiviral/immune mRNA target (*n* = 74) expression and cytokine/chemokines production (*n* = 27) was assessed on the processed biological samples (**Step 3**).

**Figure 2 cells-10-01788-f002:**
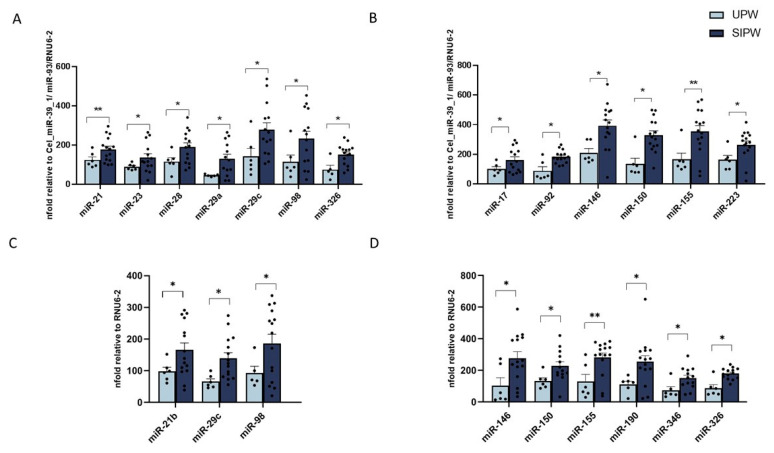
MiRNA expression analysis in plasma and placenta biopsies by PCR array. Analyses of miRNA with antiviral (**A**,**C**) and with immunological function (**B**,**D**) was performed on plasma (**A**,**B**) and placenta biopsies (**C**,**D**) of 6 UPW (light blue bars) and 15 SIPW (blue bars). Results were calculated relative to the arithmetical mean of the references available in the arrays (Cel_miR-39_1, miR-93 and RNU6-2 for plasma and RNU6-2 for placenta biopsies). Values are mean ± SEM. Significance is indicated as follows: * = *p* < 0.05 and ** = *p* < 0.01.

**Figure 3 cells-10-01788-f003:**
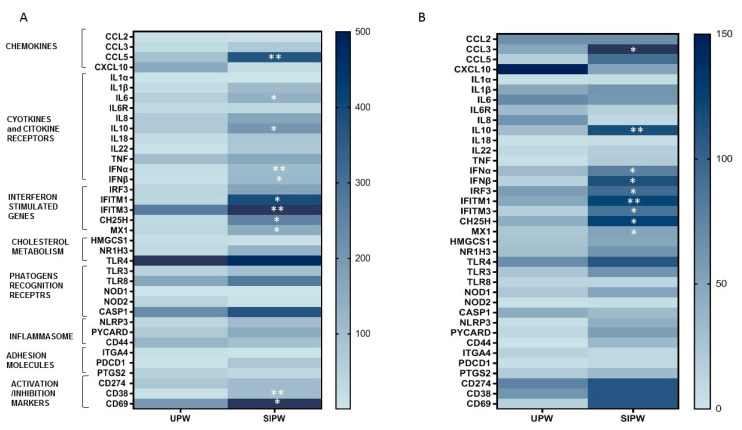
mRNA expression of genes involved in the antiviral/immune response. Quantigene Plex Gene expression technology was applied to quantify gene expression on RNA extracted from PBMCs (**A**) and placenta biopsies (**B**) isolated from 6 UPW and 15 SIPW. Gene expression (mean values) is shown as a color scale from light blue to blue (Heatmap). Only statistically significant *p* values from *t*-test comparison between UPW and SIPW are shown, as follows: * = *p* < 0.05 and ** = *p* < 0.01.

**Figure 4 cells-10-01788-f004:**
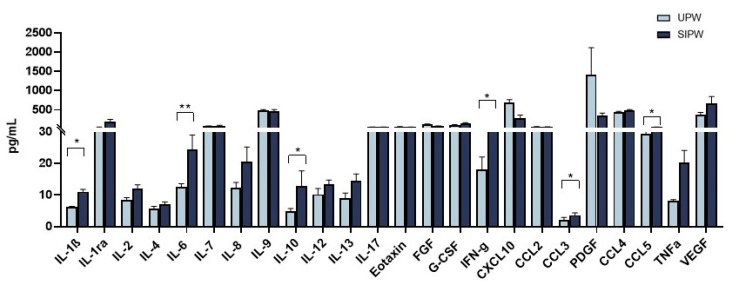
Plasma secretion of cytokines/chemokines that are part of the inflammatory response. The production of 27 cytokines/chemokines regulating immune response was assessed by Luminex assay in plasma of 6 UPW (light blue bars) and 15 SIPW (blue bars). Values are mean ± SEM. Significance is indicated, as follows: * = *p* < 0.05 and ** = *p* < 0.01.

**Table 1 cells-10-01788-t001:** Baseline characteristics of the study population on admission.

	SIPW (*n* = 15)	UPW (*n* = 6)
Maternal baseline characteristics		
Maternal age, years, median (range)	32 (21–39)	33.5 (28–40)
Third trimester of pregnancy, *n* (%)	15 (100)	6 (100)
Gestational age at admission, median (range)	39 (36–41)	40 (38–41)
RT-PCR assay of a maternal nasopharyngeal swab		
Positive, *n* (%)	15 (100)	0 (0)
Prepregnancy BMI, kg/m^2^, median (range)	23.4 (17.1–31.1)	22.9 (18.2–33.1)
Smoking habit, *n* (%)	0 (0)	0 (0)
Ethnicity, Caucasian, *n* (%)	9 (60)	6 (100)
Chronic comorbidity, *n* (%)	4 (26.6)	0 (0)
Nulliparous *n* (%)	6 (40)	1 (16.6)
Oxygen support without ICU admission, *n* (%)	2 (13.3)	0 (0)
Positive chest X-ray, *n* (%)	5 (33.3)	0 (0)
Severe case, *n* (%)	2 (13.3)	0 (0)
Admission to ICU, *n* (%)	1 (6.7)	0 (0)

**Table 2 cells-10-01788-t002:** Maternal and pregnancy outcomes in the study population.

	SIPW (*n* = 15)	UPW (*n* = 6)
Total of deliveries, *n* (%)	15 (100)	6 (100)
Delivery mode		
Vaginal, *n* (%)	10 (66.7)	3 (50)
Caesarean section, *n* (%)	5 (33.3)	3 (50)
GA at delivery, weeks median (range)	38 (36–40)	40 (38–41)
Caesarean section for severe maternal illness related to COVID-19, *n* (%)	2 (13.3)	0 (0)
Birth weight, g, median (range)	3160 (2665–3775)	2955 (2715–3500)
Umbilical artery pH, median (range)	7.34 (7.24–7.53)	7.31 (7.19–7.36)
APGAR score 5′ < 7, *n* (%)	1 (6.7)	1 (16.7)
Infected neonates, positive, *n* (%)	0 (0)	NA
NICU admission, *n* (%)	0 (0)	NA

NS: not applicable.

## Data Availability

Data sharing is not applicable to this article.

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
