# Peer review of "MiRNA Profiling in Plasma and Placenta of SARS-CoV-2-Infected Pregnant Women"

_cells, 2021, doi:10.3390/cells10071788_

Round 1

Reviewer 1 Report

This paper examines the impact of SARS-CoV-2 infection in pregnancy on circulating and placental micro RNA, mRNA, and cytokines.

This is an important study that absolutely needs to be done. I appreciate the effort the authors made and I applaud them for doing this.

However the small number of cases and methodologic problems sharply limit the value of this study in its present form.

The following details should be reported / addressed:

  • How was SARS-CoV-2 infection diagnosed? Which cases were identified due to symptoms and which due to screening?
  • For controls, SARS-CoV-2 IgG testing could be helpful for identifying whether controls were truly negative.
  • Time between infection (or diagnosis) of SARS-CoV-2 and delivery should be reported.
  • Report time between delivery and specimen collection – major impact on RNA stability.
  • Provide more detail on placental biopsies – full thickness? Villous core? Decidua and chorionic plate deliberately removed?
  • Report rate of clinical and histologic chorioamnionitis (if placentas examined histologically)
  • Are contemporary white blood cell differentials available?

These details suggest problems with the analyses.

Ideally, miRNA, mRNA, and cytokine levels would be compared using stratified or multivariate regressions to include the following covariates: Timing of infection, severity of infection, route of delivery, time between delivery and sampling, and presence of clinical or histologic chorioamnionitis.

White blood cell differential counts would help interpret the PBMC mRNA levels to rule out that expression changes are merely due to different population composition.

The authors should report the impact of these considered individually (perhaps in a supplementary table) and perform selected informative subgroup studies.

Statistical issues include:

The study would be significantly strengthened by adding patients and samples. If this is infeasible, it may be preferable to label it as a pilot or preliminary results. If the authors feel the sample size is sufficient to fully address the research question, they should make and defend that argument.

Authors should report whether their miRNA/mRNA/cytokine values are normally distributed and either transform them or use a non-parametric test.

Authors should report a Bonferroni or false discovery rate threshold and preferentially discuss those analytes that cross this threshold.

Minor wording issues and typos:

“immunologic fence” in line 42 is an unfamiliar term

miRNAs are usually described as noncoding, rather than “uncoding” line 46

My impression is that miRNA primarily act via interfering with translation, while siRNA drives degradation (line 47)

Extra space in line 52

“nearly 68 showed the same trend” is somewhat confusing. I think the authors mean “68 showed no significant difference between the groups” or similar (line 165)

Figures 2,3,4 all state “* = P<0.05 and ** = P < 0.05”. For my review, I have assumed ** = P < 0.01.

The authors should take care with claims that this is the first work of its kind. Some journals require proof of a literature search to demonstrate novelty.

“peculiar” is an odd word choice (line 289). I would say “… coupled with an immune profile in SIPW, characterized by…”

Author Response

Reviewer 1

Comments and Suggestions for Authors

This paper examines the impact of SARS-CoV-2 infection in pregnancy on circulating and placental micro RNA, mRNA, and cytokines.

This is an important study that absolutely needs to be done. I appreciate the effort the authors made and I applaud them for doing this.

However, the small number of cases and methodologic problems sharply limit the value of this study in its present form.

The following details should be reported / addressed:

How was SARS-CoV-2 infection diagnosed? Which cases were identified due to symptoms and which due to screening?

All syntomatic or asyntomatic patients were swabbed for SARS –CoV-2 before being admitted to the hospital. Patients were considered with a COVID-19 infection diagnosis defined by a positive result on a reverse-transcriptase–polymerase-chain-reaction (RT-PCR) assay of a maternal nasopharyngeal swab specimen. Nasopharyngeal swabs were obtained following the US Centers for Disease Control and Prevention (CDC) guidelines. All women underwent clinical evaluation of symptoms, laboratory analysis and radiological chest assessment at admission.

The severity degree of the infection was defined as follows. Asymptomatic: no symptoms (with negative imaging when performed); mild: one or more symptoms among fever, cough, pharyngeal pain, headache, myalgia, nausea, emesis, diarrhoea, anosmia, ageusia, but no dyspnea (abnormal imaging when performed); moderate: evidence of lower respiratory disease by clinical assessment or imaging and Oxygen saturation (SaOâ‚‚) ≥ _94% on room air; severe: one or more symptoms among SaOâ‚‚ _≤ _94% in ambient air, PaOâ‚‚/FiOâ‚‚ _<300 mmHg (i.e. arterial oxygen partial pressure/fraction of inspired oxygen), respiratory rate >30/minute or pneumonia involving more than 50% of the lungs’ _volume at X-ray scan.

For controls, SARS-CoV-2 IgG testing could be helpful for identifying whether controls were truly negative.

All the patients considered negative had never had any symptoms, no proximity to infected people and negative swabs for SARS-CoV- 2. However, according to the reviewer suggestion, all plasma samples collected from UPW were tested for the presence of SARS-CoV-2 neutralizing antibodies by an in vitro neutralization assay. Results confirmed that no UPW were infected by SARS-CoV-2. This information has been reported in the manuscript (lane 87-89): “An in vitro SARS-CoV-2 neutralization assay confirmed that no UPW was infected by SARS-CoV-2 (data not shown).”

Time between infection (or diagnosis) of SARS-CoV-2 and delivery should be reported.

According to the reviewer request, a table describing the time between diagnosis and delivery has been reported as supplementary Table 1 (Lane 155).

Report time between delivery and specimen collection – major impact on RNA stability.

We agree with the reviewer on the importance of collecting and processing the biological samples as promptly as possible to preserve RNA stability. Indeed, such information has been reported in both the graphical abstract (lane 77-80) and the method section (lane 102-109). To further, strengthen the prominence of this step the following sentence was reported in the manuscript (lane 102-107): “Few hours before delivery, 10ml maternal blood sample in EDTA was collected while full-thickness placental biopsies were obtained soon after delivery. All biopsies were ob-tained in a sterile way by a dedicated operator, after delicate washing with a sterile physi-ologist to remove any maternal blood. In order to preserve RNA stability, biological sam-ples from obstetrics and gynaecology units were immediately conveyed to the laboratory of immunology, University of Milan, to be readily processed or stored.”.

Provide more detail on placental biopsies – full thickness? Villous core? Decidua and chorionic plate deliberately removed?

As for the placental biopsies we performed a full-thickness placental immediately after delivery.

All biopsies were obtained in a sterile way by a dedicated operator, after delicate washing with a sterile physiologist to remove any maternal blood (Lane 102-107)

Report rate of clinical and histologic chorioamnionitis (if placentas examined histologically)

All the placenta, both from cases and controls, were stored and analysed at the Pathology Unit in L. Sacco Hospital. We did not observe relevant differences in placenta histopathological patterns between SIPW and UPW. We have recently published a wider study on this issue in Placenta (Placenta 110 (2021) 9–15).

Are contemporary white blood cell differentials available?

Following the reviewer request, we analysed routinely performed white blood cell counts just before delivery. Notably, despite almost all enrolled SIPW displayed mild/moderate symptoms, the lymphocyte’s cell count was significantly lower in SIPW compared to UPW. Certainly, the differences observed in miRNA/mRNA expression levels between the two analysed groups could be ascribed to such discrepancy. However, in our opinion, the overall message provided by this study, (“SARS-CoV-2 infection alters miRNA/mRNA expression levels in SIPW”) would not be modified. Definitely, this unexpected result has been introduced and commented in the manuscript in both the results section (Notably, by analysing the white blood cell count, routinely performed just before delivery (supplementary Table 1), we observed that lymphocyte’s cell count was significantly lower in SWIP compared to UPW (data not shown))” (Lane 154-157); and in the discussion ( “Moreover, we cannot exclude that the differences reported in miRNA/mRNA expression level distinguishing SIPW from UPW rely on the reduced lymphocyte’s cell count of SIPW9. ) (https://doi.org/10.1038/s41392-020-0148-4.)” (Lane 328-333)

These details suggest problems with the analyses.

Ideally, miRNA, mRNA, and cytokine levels would be compared using stratified or multivariate regressions to include the following covariates: Timing of infection, severity of infection, route of delivery, time between delivery and sampling, and presence of clinical or histologic chorioamnionitis.

As required by the reviewer we re-analysed all the targets (miRNA, mRNA and cytokine levels) by using a stratified multivariate regression analyses including timing of infection, severity of infection and route of delivery as covariates. Time between delivery and sampling as well as presence of clinical or histologic chorioamnionitis where not included in the analyses because, as previously reported, all placenta were sampled immediately after delivery, and no relevant differences in placenta histopathological patterns were observed between SIPW and UPW. Following this analyses almost all biomarkers were confirmed to be differently expressed in SIPW compared to UPW and not to be dependent on the considered confounding factors. Those targets whose significance was correlated to the included co-variates were removed from both the manuscript and figures.

White blood cell differential counts would help interpret the PBMC mRNA levels to rule out that expression changes are merely due to different population composition. The authors should report the impact of these considered individually (perhaps in a supplementary table) and perform selected informative subgroup studies.

Even if the authors agree with the importance of verifying the contribution exerted by each subcellular population to mRNA levels, this was not the scope of this study. As a matter of fact although cell separations might permit one to identify the relevant miRNA/mRNA synthetized by different cell types, it is also possible that cell interactions may be important or essential for generating the relevant miRNA/mRNA profile and a mixed cell culture more accurately reflects what might happen in vivo. Nevertheless, in the light of these promising data we intend to replicate the study on isolated cellular subpopulations to identify which one is mainly responsible for defining SIPW profile.

However, following the reviewer suggestion we analysed the results of the blood count routinely performed just before delivery and the results are summarized in supplementary table 1 and commented in the manuscript in both the results section (Notably, by analysing the white blood cell count, routinely performed just before delivery (supplementary Table 1), we observed that lymphocyte’s cell count was significantly lower in SWIP compared to UPW (data not shown))” (Lane 154-157); and in the discussion ( “Moreover, we cannot exclude that the differences reported in miRNA/mRNA expression level distinguishing SIPW from UPW rely on the reduced lymphocyte’s cell count of SIPW9. ) (https://doi.org/10.1038/s41392-020-0148-4.)” (Lane 328-333).

Statistical issues include:

The study would be significantly strengthened by adding patients and samples. If this is infeasible, it may be preferable to label it as a pilot or preliminary results. If the authors feel the sample size is sufficient to fully address the research question, they should make and defend that argument.

The observation of the reviewer is pertinent; indeed, this caveat was reported in the discussion (lane 319). However, given the uniqueness of this cohort, the complexity of coordinating sample collection and their processing to avoid RNA degradation we believe that the sample size is sufficient to address the research question. Nonetheless, according to the reviewer recommendation we specified in the manuscript that these results are preliminary and need further investigations in larger cohorts (lane 321, 322).

Authors should report whether their miRNA/mRNA/cytokine values are normally distributed and either transform them or use a non-parametric test.

Following the reviwer suggestion all biomarkers were compared between SIPW and UPW by using Students t-test or Mann-Whitney U test for continuous variables. After performing log transformation of continuous variables to approximate to normal distribution, multivariate linear regression models were performed to investigate associations between maternal infection and the study variables concentrations, taking into account the following confounding factors: timing of infection, severity of infection and route of delivery 

Minor wording issues and typos:

  1. “immunologic fence” in line 42 is an unfamiliar term

The term “fence” has been substituted with “barrier”

  1. miRNAs are usually described as noncoding, rather than “uncoding” line 46

The reviewer is right, therefore “uncoding” has been changed in “noncoding”

  1. My impression is that miRNA primarily act via interfering with translation, while siRNA drives degradation (line 47)

As both RNA degradation and interference with translation have been reported as plausible mechanisms exerted by miRNA this information has been added in the manuscript (lane 47)

  1. Extra space in line 52

Done

  1. “nearly 68 showed the same trend” is somewhat confusing. I think the authors mean “68 showed no significant difference between the groups” or similar (line 165)

The sentence has been modified according to the reviewer suggestion

  1. Figures 2,3,4 all state “* = P<0.05 and ** = P < 0.05”. For my review, I have assumed ** = P < 0.01.

We do apologize for the mistake and modified the figure legends

  1. The authors should take care with claims that this is the first work of its kind. Some journals require proof of a literature search to demonstrate novelty.

We do share the observation of the reviewer, but no other studies on this topic were so far reported in the literature. Despite recognizing the limit of the sample size we do not feel that the expression “first study” is inappropriate.

  1. “peculiar” is an odd word choice (line 289). I would say “… coupled with an immune profile in SIPW, characterized by…”

The sentence was revised according to the reviewer counsel

Reviewer 2 Report

The manuscript by Saulle et al. is intended to study the antiviral immune response in the pregnant women contract SARS-Co-V-2. The authors surveyed the expression levels of selected gene sets, which encode 84 miRNAs and 74 proteins that are involved in antiviral immune response, in 6 uninfected pregnant women (UPW) and 15 SARS-CoV-2-infected pregnant women (SIPW). To this end, PCR amplification- or RNA hybridization-based approaches were used to quantitate the RNA transcript levels of miRNAs and genes of interest in the plasma and placentas of UPW and SIPW. In addition, immunoassays were performed to measure the plasma levels of cytokines and chemokines of interest. As expected, differentially expressed miRNAs were detected in SIPW compared with UPW─16 and 9 miRNAs in plasma and placenta samples, respectively. Many of the miRNAs are shared in the two anatomical districts. On the other hand, genes for activation markers and pro-inflammatory cytokines and chemokines were upregulated in SIPW. Of note, a significant increase in the expression of host antiviral effector genes such as MX1, IFITM1, FITM3, and CH25H, was detected in the placentas of SIPW. The plasma levels of pro-inflammatory cytokines and chemokines, including IL-1b, IL-6, IL-10, IFNg, TNFa, CCL3, and CCL5, were increased in plasma from SIPW. Overall, this pilot study has identified antiviral and immunological factors in the pregnant women in response to SARS-CoV-2 infection.

Specific comments

  1. The is no information about pathological examination of the placentas from SIPW. A comparison between UPW and SIPW placentas by histopathology will further support the gene expression profiling results in the present study.
  2. The numbering in the Y-axis of the plot in Figure 4 is not clearly presented.
  3. The is no mechanistic investigation of the identified differentially expressed miRNAs and genes to better understand the pathogenesis of SARS-Co-V-2 in the placenta during pregnancy. Nevertheless, the authors have provided sufficient information in the Discussion section about the possible functions of these miRNAs and genes in response to SARS-Co-V-2 infection.  Please provide information about the pathological examination of SWIP placentas in your revised manuscript.

Author Response

The manuscript by Saulle et al. is intended to study the antiviral immune response in the pregnant women contract SARS-Co-V-2. The authors surveyed the expression levels of selected gene sets, which encode 84 miRNAs and 74 proteins that are involved in antiviral immune response, in 6 uninfected pregnant women (UPW) and 15 SARS-CoV-2-infected pregnant women (SIPW). To this end, PCR amplification- or RNA hybridization-based approaches were used to quantitate the RNA transcript levels of miRNAs and genes of interest in the plasma and placentas of UPW and SIPW. In addition, immunoassays were performed to measure the plasma levels of cytokines and chemokines of interest. As expected, differentially expressed miRNAs were detected in SIPW compared with UPW─16 and 9 miRNAs in plasma and placenta samples, respectively. Many of the miRNAs are shared in the two anatomical districts. On the other hand, genes for activation markers and pro-inflammatory cytokines and chemokines were upregulated in SIPW. Of note, a significant increase in the expression of host antiviral effector genes such as MX1, IFITM1, FITM3, and CH25H, was detected in the placentas of SIPW. The plasma levels of pro-inflammatory cytokines and chemokines, including IL-1b, IL-6, IL-10, IFNg, TNFa, CCL3, and CCL5, were increased in plasma from SIPW. Overall, this pilot study has identified antiviral and immunological factors in the pregnant women in response to SARS-CoV-2 infection.

Specific comments

  1. The is no information about pathological examination of the placentas from SIPW. A comparison between UPW and SIPW placentas by histopathology will further support the gene expression profiling results in the present study.

All the placenta, both from cases and controls, were stored and analyzed at the Pathology Unit in L. Sacco Hospital. We did not observe relevant differences in placenta histopathological patterns between SIPW and UPW. We have already published a wider study on this issue in Placenta (Placenta 110 (2021) 9–15)

  1. The numbering in the Y-axis of the plot in Figure 4 is not clearly presented.

According to the reviewer suggestion the Y-axis of the plot in Figure 4 has been modified

  1. There is no mechanistic investigation of the identified differentially expressed miRNAs and genes to better understand the pathogenesis of SARS-Co-V-2 in the placenta during pregnancy. Nevertheless, the authors have provided sufficient information in the Discussion section about the possible functions of these miRNAs and genes in response to SARS-CoV-2 infection. Please provide information about the pathological examination of SWIP placentas in your revised manuscript.

The observation of the reviewer is appropriate and this limitation was reported in the discussion (lane 320). Indeed, this was an exploratory study aimed at profiling miRNA/mRNA expression in biological samples from SIPW. However, based on these promising results we intend to further investigate the mechanistic relationship associating miRNA and mRNA expression.

Information concerning the pathological examination of SWIP placentas have been reported in the revised version of the manuscript: (Lane161-163)

Round 2

Reviewer 2 Report

The authors have addressed most of the concerns I had with the previous submission.